# Endocrine Disruptors and Prostate Cancer

**DOI:** 10.3390/ijms23031216

**Published:** 2022-01-21

**Authors:** Margherita Corti, Stefano Lorenzetti, Alessandro Ubaldi, Romano Zilli, Daniele Marcoccia

**Affiliations:** 1Istituto Zooprofilattico Sperimentale del Lazio e della Toscana, Via Appia Nuova 1411, 00178 Rome, Italy; margherita.corti@izslt.it (M.C.); alessandro.ubaldi@izslt.it (A.U.); romano.zilli@izslt.it (R.Z.); 2Department of Food Safety, Nutrition and Veterinary Public Health, Istituto Superiore di Sanità (ISS), 00161 Rome, Italy; stefano.lorenzetti@iss.it

**Keywords:** endocrine disruptors, prostate cancer, nuclear androgen receptor, membrane androgen receptors

## Abstract

The role of endocrine disruptors (EDs) in the human prostate gland is an overlooked issue even though the prostate is essential for male fertility. From experimental models, it is known that EDs can influence several molecular mechanisms involved in prostate homeostasis and diseases, including prostate cancer (PCa), one of the most common cancers in the male, whose onset and progression is characterized by the deregulation of several cellular pathways including androgen receptor (AR) signaling. The prostate gland essentiality relies on its function to produce and secrete the prostatic fluid, a component of the seminal fluid, needed to keep alive and functional sperms upon ejaculation. In physiological condition, in the prostate epithelium the more-active androgen, the 5α-dihydrotestosterone (DHT), formed from testosterone (T) by the 5α-reductase enzyme (SRD5A), binds to AR and, upon homodimerization and nuclear translocation, recognizes the promoter of target genes modulating them. In pathological conditions, AR mutations and/or less specific AR binding by ligands modulate differently targeted genes leading to an altered regulation of cell proliferation and triggering PCa onset and development. EDs acting on the AR-dependent signaling within the prostate gland can contribute to the PCa onset and to exacerbating its development.

## 1. Introduction

The prostate is the main accessory gland of the male reproductive system, which plays an essential role in male reproduction. The prostatic fluid, secreted by the epithelium of the gland, contains essential molecules contributing not only to the prostate functionality but also to the steps associated with ejaculation and hence to male fertility, such as sperm motility and capacitation [1,2]. Among them are proteins of the kallikrein family (e.g., the prostate-specific antigen PSA or kallikrein 3/KLK3), and other essential factors such as the trace element zinc or the Krebs cycle intermediate [1,2]. An important role is played by PSA, a cysteine proteinase normally produced by the prostatic epithelial cells. PSA is secreted apically into the ductal lumina, where it cleaves semenogelins I and II, the physiological substrates that mediate gel formation in semen and in the seminal coagulum; it is then removed by ejaculation [1,2,3]. A reduction of the PSA secretion could influence sperm fertilization potential; on the contrary, an increase in PSA secretion can be used as a prostate cancer (PCa) biomarker [2].

PCa is a clinically heterogeneous disease that may be either aggressive, with a clinical progression to metastasis, or asymptomatic, with an indolent course [4,5]. PCa is a common malignancy in men, and it is the fifth-highest cause of death worldwide [4]. Despite this, the mechanism underlying this high disease incidence is still not well understood. The treatment of this cancer is mainly based on surgery and radiotherapy with the exception of patients non-suitable for these treatments, who are treated with androgen ablation therapy [6].

Commonly, the diagnosis of PCa is made based on elevated plasmatic blood levels of PSA (>4 ng/mL). The androgenic hormones regulate normal prostate gland growth and function by interacting with the androgen receptor (AR), whose gene expression in PCa appears to be deregulated; this plays a central role in development and metastatic progression [7,8]. PCa is considered an androgen-dependent cancer [9,10], its growth requires androgens, such as testosterone (T) or the more potent dihydrotestosterone (DHT), and it was noted that androgen deprivation therapy (ADT) led to prostate tumor regression in up to 80% of cases [11]. Despite this, tumors eventually begin to grow, despite continued antiandrogen treatment [11]. The resistance of PCa at the ADT could depend on several pathways centered on androgen signaling, including intra-tumoral and adrenal androgen production, AR-overexpression and amplification, expression of AR mutants, and constitutively, active AR splice variants [12].

In PCa, besides the nuclear AR, four distinct proteins play a role in the PCa growth. This includes the transient receptor potential melastatin 8 (TRPM8), the oxoeicosanoid receptor 1 (OXER1), the zinc transporter ZIP9/SLC39A (namely, Zrt- and Irt-like Protein 9 or Solute Carrier family 39 member 9), and the G-protein-coupled receptor family C group 6 member A (GPRC6A). These proteins have been proposed as alternative membrane androgen receptors (mARs) that could be involved in the survival of PCa cells [13,14].

Moreover, in addition to androgens, estrogenic hormones are also involved in PCa development. Indeed, the use of anti-estrogens has been recognized to have a therapeutic role in PCa management [15,16]. Both estrogen receptors (ERs), namely ERα and ERβ, are expressed in the prostate gland during its development. In adulthood, ERα is normally found primarily in stromal cells and ERβ in differentiated epithelium [15,16,17]. It is noteworthy that the prostate gland is particularly sensitive to estrogen exposure during its critical developmental period, due to adult estrogenic responses [15,18].

The established risk factors for PCa are mainly the age and the race of the individuals. For example, African Americans are twice as likely as European individuals to develop or die from PCa [19]. Other factors, such as genetics (family history), diet, and environmental factors can also have an impact on PCa risk [20]. These also include endocrine disruptors (EDs), defined as exogenous substances or a mixture that “alters function(s) of the endocrine system, causing adverse health effects in an intact organism, or its progeny, or in a (sub)populations” [21]. EDs can act at a low dose and are found in many daily products, such as beverage and plastic bottles, food cans and overall food packaging, epoxy plastic-based electronics, recycled paper, and so on [18,22]. The main exposure to these contaminants occurs through the intake of contaminated water and food, but can also be taken through other sources, such as direct dermal contact or inhalation of polluted air [22,23]. Most EDs act similarly to hormone-like chemicals, mostly as estrogen-like or antiandrogens, because they possess a chemical structure similar to natural endogenous hormones. Thus, EDs can mimic or block different hormonal pathways [24]. In fact, EDs interact with sexual receptors, both ER and AR, and with other non-nuclear receptors [24,25].

There are several in vitro studies in human prostate epithelial cells and in vivo studies in animal models as well as epidemiologic studies that indicate a possible association between EDs and PCa risk [4,15,26,27]. However, a direct connection between a certain chemical and human PCa risk has not yet been established. Indeed, a direct association between PCa risk in humans and EDs exposures is difficult due to their peculiarities and the way of acting over time of these toxic compounds.

## 2. Sex Steroid Hormones (Androgens)

Sex hormones are derived from cholesterol and belong to a class of chemical compounds known as steroid hormones. These classes of hormones are very important for male sexual development, reproduction and health and metabolic homeostasis. This is especially true for androgens [28]. The major naturally occurring steroids with androgenic activity are (in decreasing order of relative potency): 5α-dihydrotestosterone (DHT, 150–200%), testosterone (T, 100%), 3α-androstanediol (3α-diol, 65%), androst-4-ene3,17-dione (AD, 25%), androsterone (AND, 10%), and dehydroepiandrosterone (DHEA, 10%) [18] (Figure 1).

About 95% of T is produced and secreted by Leydig cells in the testis, and the remaining 5% is produced in adrenal glands by conversion of precursors (e.g., DHEA, DHEA sulfate, and androstenedione) [18]. T is converted to DHT and 17β-estradiol (E2, the main active estrogen), by the enzymes 5α-reductase (SRD5A) and aromatase, respectively [18,29]. There are three different isoforms of SRD5A, encoded by separate genes (SRD5A1, SRD5A2, and SRD5A3) and, among these, the most prevalent and biologically active isoenzyme is the SRD5A1. The major sites of distribution of SRD5A in the human tissues are both in the reproductive organs (where the isoform SRD5A2 is prevalent) and in non-reproductive tissues, such as liver and skin, in which isoform SRD5A1 is the most commonly present [18,30]. In experimental animal models, it has been observed that the up-regulation of SRD5A1 promotes the formation and development of rat prostate intraepithelial neoplasia in vivo [31].

As mentioned above, the two most important androgens in the male are T and DHT, which act by binding the same nuclear receptor AR with a different tissue specificity: each androgen has its own specific role during male sexual differentiation. Indeed, T is directly involved in the development and the differentiation of epididymides, *vasa deferentia*, seminal vesicles and ejaculatory ducts, whereas DHT is the active ligand in several other androgen-target tissues such as the prostate gland, scrotum, urethra, and penis [32]. Moreover, the two androgens interact in a different mode with the AR: T has a two-fold lower affinity compared to DHT, which in turn exhibits 10 times higher affinity to the AR than T [18].

Sex hormone-binding globulin (SHBG) determines the equilibrium between free and protein-bound androgens in blood and regulates their access to target tissues. DHT has the highest affinity ~1 nm K_d_ for SHBG compared to T, which bind with ~5 times lower affinities, when compared with DHT [33].

The plasma levels and biological actions of sex steroids is regulated not only by the bind with SHBG but also by the hypothalamic–pituitary–gonadal axis. Indeed, T production depends upon the pulsatile secretion of the luteinizing hormone (LH), an adenohypophysis gonadotropin [18,34]. LH-regulated T production and its endogenous secretion is pulsatile and diurnal, with the highest peak occurring in the morning and the lowest in the evening.

Endogenous T levels decrease in males with aging. However, at the same time, the incidence of androgen-related pathologies, such as PCa and benign prostate hyperplasia (BPH), increases with age. This increased incidence could be related to the local enzymatic conversion of T to DHT by 5α-reductase, since the up-regulation of SRD5A, the gene encoding 5α-reductase, has been shown [35,36]. Indeed, benign lesions such as BPH increase markedly in 90% of men older than 80 years because both the estrogen/testosterone intra-prostatic ratio and the ER overexpression appear to be higher when compared to that present in young males [37].

## 3. Mechanisms of Androgen Action in Prostate Gland: The Androgen Receptor

Androgens (DHT and T) exert their actions by binding to the AR, a ligand-activated transcription factor, which belongs to the third group of the nuclear receptor (NR) superfamily (NR3C4, nuclear receptor subfamily 3, group C, member 4) [38]. The AR gene is present on the X chromosome (Xq11–12) and consists of eight exons interrupted by introns of different lengths (0.7–2.6 kb). It codes for a protein of approximately 920 amino acids, whose structure is similar to the other NRs [18]. The AR protein is composed by four functional domains: the intrinsically disordered *N*-terminal domain (NTD, aa 1–558), the DNA-binding domain (DBD, aa 558–624), the *C*-terminal domain (aa 676–919) and the ligand-binding domain (LBD) (Figure 2) [18,37].

The classical nuclear (or genomic) mechanism of the androgen action presents the inactive AR protein resides in the cytoplasm associated with a chaperone complex (health shock protein HSP90). Upon hormone binding to the AR with high affinity at the level of the LBD, the receptor–chaperone complex rearranges and the AR undergoes conformational modifications. Subsequently, AR translocation to the nucleus and its homodimerization allows the ligand-bound receptor to recognize target genes, such as PSA and transmembrane protease serine 2 (TMPRSS2), via specific promoter and enhancer sequences, known as androgen response elements (AREs) [18,37]. Target gene recognition by AREs leads to the modulation of AR-target genes by induction or repression of the transcription machinery: so far, more than 300 AR-interacting proteins have been identified; among them it has been reported 168 co-activators and 89 co-repressors ([38] and refs therein). Most of these co-regulators are chromatin-modifying enzymes. Once bound to the chromatin, AR recruits numerous coregulatory proteins to modulate the AR-transcriptional complex transcription, leading to cell growth and survival responses [39,40,41].

In addition, DHT and T, similarly to some other steroid hormones, also present a non-genomic signaling/action, with a rapid response (seconds to minutes), the AR localized on the lipid membrane rafts microdomain, leading to extra-nuclear mechanisms that can directly involve extracellular signal-regulated kinase (ERK), the phosphatidyl-inositol 3-kinase (PI3K)/Akt pathway, G-protein coupled receptors (GPCRs), intracellular Ca^2+^ concentration, and cyclic adenosine monophosphate (cAMP) levels [42,43].

The sequence comparison between AR and ERs shows a similar sequence for the palmitoylation, a post-translational modification, present in both types of sex steroid receptors. The palmitoylation of a conserved motif in the ligand-binding domain is critical for the membrane localization of both ERs and AR [18,44]. The localization at the level of the plasma membrane of the AR and its interaction with caveolin-1 (Cav-1), a major component of the caveolae membrane structure, is due to palmitoylation [44,45]. Cav-1 enhances AR-transcriptional activity after the androgen binds to the AR as it may increase nuclear translocation and phosphorylation of the AR itself. On the other hand, a down-regulation of Cav-1 decreases AR membrane localization [44,45]. Overall, the effects elicited by androgens are obtained by different signal transduction pathways (e.g., nuclear and extra-nuclear), whose activation depends on the cellular context of the target cell, the AR intracellular localization (e.g., membrane-bound, cytosolic, nuclear), and the ligand itself (e.g., T vs. DHT).

## 4. Androgen Receptor Mutations and Splice Variants in the Prostate Gland

AR deregulation plays a central role in the onset and progression of PCa to an advanced metastatic disease [7,8]. The mechanism for abnormal AR activation includes somatic AR missense mutation that have been found most frequently first in the LBD and after in the NTD sequence [46,47]. These mutated AR variants—to date over 150 AR mutations have been identified in PCa tissue—usually result in small changes within the AR protein influencing the ligand-binding specificity [47]. Most of these mutations consist of a single-base substitution that rarely occurs within the germline [48]. The most frequent example is the mutation AR^T877A^ located in the AR-LBD. This mutation consists of the substitution of an alanine (Ala) with a threonine (Thr), resulting in the loss of specificity for the agonist. Therefore, the mutated AR might be activated not only by androgens but also by other steroid hormones, such as progesterone, estrogens, as well as by DHT metabolic products, or antiandrogen compounds, such as flutamide (FTA) [47,48,49,50].

Specific AR germline polymorphisms have been associated with an increased risk of developing PCa, up to six-fold higher than the general population (e.g., AR^R725L^) [51,52]. Other mutations, as such as the AR^F876L^, lead to a change in the LBD domain conferring resistance to enzalutamide (a well-known substance that inhibits the AR), whereas several other mutations increase the binding capacity of AR with co-regulators, thus increasing AR-transcriptional activity (e.g., AR^H874Y^, and AR^W435L^ mutations) [47,52]. AR gene amplification causes an overexpression of the AR protein that, in turn, increases the sensitivity of prostate cells to the lower level of androgens, in other words allowing PCa to grow even in a low-androgen environment [48,52].

A specific group of AR mutations, the AR splice variants (AR-Vs), have been identified in PCa, even if many of them have also been identified in non-cancerous tissues. Indeed, these AR isoforms are naturally occurring splice variants encoded by alternative AR transcripts. They derived from cryptic exon downstream regulation of the sequence that codes for the DBD, which presents premature stop codons: most translated AR proteins retain NTD and DBD, but lack the LBD being constitutively active [52,53]. AR-Vs play a critical role in PCa development and progression since they possess a ligand-independent transcriptional activity and are strongly up-regulated in hormone refractory PCa [54]. However, many of them present a unique pattern of target genes that differs from those regulated by full length ARs [55]. The most common AR splice variant, AR-V7, is a membrane-associated AR splice variant, which also lacks the LBD and therefore it cannot function alone as a transcription factor. Indeed, its mechanism of action is strictly extra-nuclear [5]. In particular, AR-V7 induces the expression of genes related to cell-cycle progression, including the UBE2C gene, a transcription factor that regulates the expression of genes involved in G1/S and G2/M transition and M phase progression [52]. Finally, AR-V7 overexpression has been associated with an increased risk of biochemical disease recurrence after radical prostatectomy in hormone-naïve PCa [5,52].

## 5. Membrane Androgen Receptors (mARs) and WNT-Pathway in Prostate

In recent years, it has been demonstrated that androgens can modulate several cellular processes that are important for cell growth and that in PCa progression and metastasis certain receptors localized at the plasma membrane (membrane AR or mARs) and/or impinging on Wnt-pathway are differentially expressed in comparison to healthy prostate tissues [13,49,50,56].

Four distinct proteins (TRPM8, OXER1, ZIP9 and GPRC6A) localized at the plasma membrane of prostate epithelial cells have been proposed as mARs [13]. TRPM8, OXER1, and GPRC6A activities are also modulated by endogenous ligands structurally unrelated to androgens, whereas ZIP9 appears to be activated solely by androgens [14,57].

TRPM8, a member of the melastatin-related transient receptor potential family, is a Ca^2+^-selective cation androgen-regulated channel. TRPM8 is expressed throughout the male urogenital tract and in other different tissues such as the peripheral nervous system [13,14,58,59,60,61]. The role of TRPM8 in PCa appeared to occur in early-stage prostate tumors, in which it is significantly up-regulated [13,14,59,62], whereas its expression is reduced in androgen-independent advanced stages of PCa, and it has no effect on cell migration, proliferation, and invasion [14,56,63]. TRPM8 has been recognized as an AR because of its steroid specificity and androgen-binding affinity [13]. TRPM8 is a potential PCa biomarker found in both malignant and non-malignant tissue samples. PSA is expressed at the highest level (in terms of absolute amounts of mRNA), with lower levels of human kallikrein 2 (hK2) and TRPM8. Moreover, TRPM8 expression but not PSA expression in malignant tissues results are greater than in non-malignant tissue [63]. TRPM8 is observed in the plasma membrane of PCa cells, although the protein has also been observed in intracellular membranes [13]. Loss of TRPM8 expression in metastatic PCa cells may, therefore, be associated with a reduction in PCa cell destruction [13].

OXER1 is a G-protein coupled receptor usually activated by 5-lipoxygenase metabolites of arachidonic acid, 5-oxoeicosatretraenoic acid (5-oxo-ETE) and other 5-eicosatretraenoic acid derivatives. OXER1 is expressed in several human tissues, including PC cells [13,14], and it is known to stimulate the effects of 5-eicosatretraenoic acid derivatives on steroidogenesis, inflammatory processes and cell proliferation, but also survival of PCa cells [14,64,65]. In silico studies based on molecular docking analysis have been predicted where T occupies the same binding pocket as 5-oxo-ETE on OXER1 [13,14,66]. OXER1 has been proposed as a potential therapeutic target for inflammatory diseases, as well as for PCa, because it prevents apoptosis of PCa cells [13,14,64].

GPRC6A is also a G-protein coupled receptor belonging to the class C group 6 subtype A expressed in humans [67,68]. GPRC6A is involved in the modulation of several pathophysiological signalings important in bone metabolism, interactions between bone and gonads, androgen production in PCa and Leydig cells, male fertility, PCa tumorigenesis, insulin secretion, energy metabolism, and inflammatory responses [67,69]. In PCa, GPCR6A is expressed at high levels, and it is correlated with increased tumor size, distant metastasis and tumor recurrence [14,68]. Indeed, in AR-independent PCa, GPRC6A overexpression has been demonstrated to mediate rapid, non-genomic signaling in response to androgen binding [70,71]. Computational studies have demonstrated the ability of androgens, in particular testosterone, to bind GPRC6A and in vitro it has been shown that such a binding leads to ERK1/2 phosphorylation and activation and to a decreased expression of the tumor-suppressor Early Growth Response Protein 1 [72]. Moreover, T binding to GPRC6A can lead to the activation of Akt and mammalian Target of Rapamycin (mTOR) signaling pathways resulting in an increased cell proliferation and inhibition of autophagy in PCa [73]. It is worth noting that in Asian populations the association between GPRC6A gene and PCa risk has been reported [70] and even a specific polymorphism (rs2274911) in GPRC6A has been associated with an increased PCa risk, since this mutation promotes cell proliferation and is associated with increased PSA serum levels [71].

ZIP9/SLC39A9 is a zinc-importing protein belonging to a 14-member family regulating zinc transport from the extracellular space to the cytoplasm [13,14]. ZIP9 is a dual function protein acting both as a Zn^2+^ transporter and as a mAR: this dual role is exerted through G proteins involved in apoptotic pathways activated by androgens [13,14,70]. ZIP9 has been demonstrated to possess high binding affinity with T, while other endogenous androgens such as androstenedione and DHT display low affinity [14,70]. ZIP9 has been demonstrated to be highly expressed and bound by androgens in ovarian, breast, and PCa cells, the same hormone-sensitive tumors in which testosterone has been demonstrated to increase intracellular Zn^2+^, where it led to high Zn^2+^ concentration-mediated apoptosis [13]. Finally, ZIP9 has been shown to mediate a testosterone-induced, AR-independent increase of cell migration in metastatic PCa cells [74].

Recent studies have shown that dysregulation of the Wnt signaling pathway plays an important role in PCa progression [56,75]. Wnt proteins are essential in the regulation of embryonic and organ development and in cell proliferation, migration, as well as differentiation. There are 19 human Wnts regulating several pathways associated with multiple cell surface receptors, as such as transmembrane frizzled (FZD) receptors, low-density lipoprotein receptor (LRP)4/5/6, receptor tyrosine kinase-like orphan receptor (ROR)1/2, and receptor-like tyrosine kinase RYK (RYK) that leads to the activation of the canonical (β-catenin-dependent) and non-canonical (β-catenin-independent) pathways [56,75]. The Wnt signaling pathway modulates the regulation of AR expression mRNA expression by TCF transcription factors. It represents a process through which Wnt signaling can modulate the activity of the androgen signaling pathway in prostate and PCa cells [56]. The transcriptional potential of the TCF family transcription factors are activated by the bond with β-catenin. Moreover, β-catenin can also directly bind to the ligand-engaged AR protein [76,77,78,79,80,81] and this binding promotes an effect on the ability of the AR to induce transcription of androgen-sensitive gene products. Indeed β-catenin (wild-type or mutated) is considered a ligand-dependent co-activator of AR transcription [82]. Furthermore, binding β-catenin the AR promotes the movement of the latter into the nucleus. There, β-catenin protein remains complexed to AR protein at its binding site on chromatin, which is associated with an androgen-regulated element within the promoter region of the human PSA gene [83]. The interaction between Wnt and AR has been demonstrated to involve another critical molecule in the Wnt signaling pathway: GSK-3β [84]. Indeed, in PCa cells the overexpression of GSK-3β has been associated with a reduced expression of PSA [85]. The Wnt signaling pathway has been proven to regulate the expression of genes that are targets of the LEF-1/TCF family of transcription factors that bind to the AR promoter [86]. TCF has been associated with being able to bind the site region of DNA upstream of the transcription start site of the human AR gene [87]. In PCa cells, it has been demonstrated that the TCF-binding sites were associated with β-catenin, and this TCF-binding site for both for β-catenin and AR has been observed on the PSA promoter [88]. Therefore, Wnt and androgen signaling pathways likely have numerous consequences on the development, growth, and progression of PCa.

## 6. Pesticides/Biocides and Plasticizers

Epidemiologic studies have linked farming to an increased PCa risk [89,90,91]. Indeed, in these studies among farmers a significant excess of both PCa incidence and mortality has been revealed among pesticide applicators compared to the general population [92]. The exposure to certain pesticides, biocides and plasticizers (Figure 3) have been associated in several studies with increased PCa [15,93].

Human exposure to pesticides may occur through occupational exposure in the case of agricultural workers or through non-occupational pathways for residents living close to agricultural lands and for bystanders. The same type of direct or indirect exposure occurs with those who work with biocidal products. Instead, the exposure of the general population to pesticides occurs mainly through diet either eating food or drinking water contaminated with pesticides. Non-occupational exposure originating from pesticide residues (MRLs) in food, air, and drinking water generally involves lower doses. This type of exposure can be considered chronic (or semi-chronic). Pesticides/biocides have been defined as substances or mixtures of substances intended for controlling, preventing, destroying, repelling, or attracting any biological organism deemed to be a pest. There are substances that can be considered both pesticides and biocides but in general the use of pesticides is specific to crops or plants overall, whereas biocides can be used in all other fields. Insecticides, herbicides, defoliants, desiccants, fungicides, nematocides, avicides, rodenticides, and hospital disinfectant (e.g., biocides) are some of the many types of existing pesticides and biocides [94]. Plasticizers have been classified as additives that increase the plasticity or the viscosity of a material. Among the plasticizers commonly used in food packaging (e.g., plastic containers) and in medical devices (e.g., blood storage bags and intravenous delivery systems) di(2-ethylhexyl) phthalate (DEHP), di-isononyl phthalate (DINP), dibutyl phthalate (DBP), and bisphenol A (BPA) can be found [95]. Several reports on the NR antagonist activities of pesticides/biocides based on either NR direct binding or different transactivation assays demonstrated that these substances (or their metabolites) have an antiandrogen-like activity via AR binding [18], whereas plasticizers act via different NRs. Some substances commonly used as pesticides/biocides and plasticizers are known to act as EDs as well.

Below, some chemicals of interest with known endocrine interference activity on the prostate gland representative of pesticides, biocides and plasticizers will be discussed. According to an Italian report on the sales and use of fungicide and neonicotinoids, vinclozolin (VIN) and imidacloprid (IMI) were selected based on their high use [96]. Among pyrethroid molecules, cypermethrin (CYPE) was also selected due to its wide use in agriculture, veterinary, and medical applications as insecticide and biocide. Finally, for their wide use as additives (plasticizers) in the plastics industry, food packaging and food contact materials, the phthalates DEHP and DBP and the bisphenol BPA have been selected for their well-known characteristics of EDs [97,98,99].

Vinclozolin (VIN) is a dicarboxymide fungicide widely used on fruit and vegetables. It has been demonstrated that it acts as an AR antagonist in vitro and/or in vivo leading to an interference with the action of androgens in developing, pubertal, and adult male rats [100,101,102]. The mode of action of VIN has been associated with AR antagonism and, for this reason, there were no reported associations between this compound and PCa, an androgen-dependent disease [15]. Exposure to VIN during the critical period of sexual differentiation results in sexual abnormalities expressed later in the adult male rat [15]. Indeed, VIN exposure during fetal gonad sex differentiation has been demonstrated to alter the epigenetic programming of the germline. It has also been demonstrated in an in vivo study that after exposure of gestating female rats to VIN, prostate epithelial and stromal cells from young VIN lineage animals have been shown to have changes in DNA methylation and ncRNA expression, as well as in mRNA gene expression [103]. With the same molecular mechanism and with almost the same potency as the classical antiandrogen drug flutamide, the two VIN primary metabolites, 2-[[(3,5-dichlorophenyl)-carbamoyl]oxy]-2-methyl-3-butenoic acid (M1) and 3′5′-dichloro-2-hydroxy-2-methylbut-3-enanilide (M2), competitively inhibit the bind of androgens with the human AR and consequently the expression of androgen-target genes [104,105]. The metabolites M1 and M2 have been demonstrated to bind to the AR and to acts to act as antiandrogens, competitively inhibiting the bind of androgens with AR, which leads to an inhibition of androgen-dependent gene expression in vitro and in vivo [101,102,103,104,105,106,107]. In vitro studies have shown that VIN can affect prostate epithelial cells: it has been demonstrated that it is able to decrease DHT-induced PSA secretion and to decrease AR nuclear accumulation and its phosphorylation [92], thus impairing the conformational changes necessary to induce the AR-mediated transcriptional activation modulated by the AF-1 region [108]. A further antiandrogen-like effect of VIN is on DHT-induced SRD5A1 gene expression in PCa cells. An interesting recent in vitro study demonstrated that VIN and its metabolites can interact with the mARs, in particular binding to ZIP9 blocks T actions, suggesting that androgen-like functions mediated by ZIP9 can be also susceptible to disruption by pesticides in an antiandrogen manner [13]. In several recent in vivo studies, the exposure to VIN during the period of gestation induces stromal and epithelial cell changes in both the epigenome and transcriptome that lead to prostate disease susceptibility [108,109].

Cypermethrin (CYPE) is a pyrethroid broad spectrum commonly used as insecticide. Consumption of food containing pyrethroid residues, particularly vegetables and fruit, is the primary route of exposure [110]. This substance may induce reproductive damage in males, such as testicular lesions, decreased sperm count, motility changes, and morphologic abnormalities [111]. Moreover, it has been demonstrated that CYPE acts as an ED with antiandrogen-like activity responsible for male reproductive [112]. The antiandrogen-like action of CYPE has been involved in CYPE-induced male reproductive toxicity by targeting the AR signaling pathway to inhibit the AR-transcriptional activity. Indeed, CYPE has been associated with inhibition of the AR transcription disrupting the interaction between the AR and the co-activator steroid receptor co-activator-1 (SRC-1) [113]. In PCa cell lines, the AR antagonism of CYPE on the co-regulators in IL-6-iducend AR signaling pathway has been also demonstrated. In fact, CYPE treatment has been shown to inhibit the PCa cell line growth induced by IL-6, because of inhibition of recruitment of SRC-1 and SMRT to the AR mediated by IL-6 involved in the mechanism of CYPE functioning as an AR antagonist [114]. Moreover, the antiandrogen-like activity of CYPE has been demonstrated to inhibit AR transcription, impairing the recruitment of two AR co-activators ARA70 and ARA55 [112]. The insecticide CYPE can also inhibit epidermal growth factor receptor (EGFR) activity and the downstream MAPK activation by interfering with non-classical T signaling in Sertoli cells, leading to reduced cell viability and proliferation in PCa [114]. Moreover, the association between the genetic variation present on chromosome 8q24 and the risk of PCa has also been demonstrated and has been modified by the exposure to pyrethroids [114].

Imidacloprid (IMI) is a neonicotinoid widely used due to its selective poisonousness to insects [115]. IMI was authorized in 1991 and it represents more of the 40% of the total use of all neonicotinoid insecticides [116]. Indeed, IMI has been detected in the human body according to several studies [117,118,119] due to its extensive use that leads non-target organisms to be highly exposed to its residues. The toxicity of IMI has been linked to hepatotoxicity and nephrotoxicity, as well as neurotoxicity in several experimental models [120,121]. Moreover, IMI has also been demonstrated be very toxic in human prostate epithelial cells and able to induce apoptotic effects as well as oxidative stress [122]. In in vivo study, the IMI exposure has been demonstrated to induce weight changes in absolute and relative prostate weights such that it could always be associated with a decrease in T [123,124]. Therefore, IMI has been shown to be associated with EDs [105]. In Mikolić and Karačonji [116], it has been shown that IMI interacts with AR in a still unclear mechanism. Indeed, IMI could significantly inhibit the expression of AR in the testicular tissue, indicating a possible reduction of the function of AR [125]. In vitro molecular docking has been demonstrated where IMI docks into the active site of AR and three key hydrogen bonds are formed with active site residues Glu11, Gln41 and Lys138. The AR-IMI complex showed a binding free energy value that suggest a stable binding between IMI and AR. Moreover, in vivo study IMI has been demonstrated to act negatively on the male reproductive function through alteration of testosterone levels and sperm parameters [126].

Di(2-ethylhexyl) (DEHP) and dibutyl phthalate (DBP) are commonly used as plasticizers. Plasticizers are particular substances that increase the plasticity or viscosity of a material and are commonly used in food packaging and in medical devices [127]. In in vivo animal models, DBP has been demonstrated to negatively affect the reproductive system, in particular sperm morphology, sperm count, sperm motility as well as the weight of testis, prostate and seminal vesicle [127]. It has been also shown [128] that in vivo exposure to a phthalate mixtures during the critical window of prostate development may lead to an alteration in the sex reproductive system and in the microRNA (miRNome) and transcriptome profiles [128]. Perinatal exposure to phthalate mixtures resulted in a prostate weight variation and induced the deregulation in mRNA and miRNA prostate expression profile, mainly in those associated with both miR-141-3p and miR-184 up-regulation [128]. Moreover, it has been demonstrated in vivo that the phthalate DBP leads to a reduced expression of Bone Morphogenetic Protein type 4 (BMP-4), known to regulate both cell proliferation and androgen level [129].

Although the phthalate DEHP does not bind to AR, in utero exposure to phthalates disrupts the differentiation program of androgen-dependent tissues in male rat offspring [130,131,132,133,134]. Indeed, in utero exposure to both DEHP and DBP has been shown to affect reproductive tissues, thus leading eventually to the onset in adults of endocrine-related diseases. Taking into account the role of phthalates, as other EDs, to interfere also with the processes of epigenetic inheritance of adult-onset diseases by modulating the epimutations of DNA methylation in reproductive cells, it is also tempting to speculate that such molecules could affect and reprogram all developmental aspects of sex-specific tissues [135].

An interesting recent in silico docking study has characterized the bond of DEHP and its metabolites with AR to predict the potential endocrine-disrupting effects of these metabolites in AR signaling [136]. The data have shown that DEHP and its metabolites interacted with the ligand-binding pocket of AR, forming amino-acid residue interactions, hydrogen bonding, and *pi–pi* interactions. This binding energy of DEHP with AR has been shown to be similar to that of native ligand T, and the amino-acid residue interactions of DEHP metabolites had 91–100% similarity compared to that of T [136]. Therefore, considering the structural binding data, the potential role of DEHP metabolites to disrupt AR signaling has been suggested, which may lead to androgen-related reproductive dysfunction. In a recent in silico study, the DBP protein target was evaluated to investigate the AR–ligand interaction, confirming how such phthalate activates, independently from the receptor AR, the androgen signaling pathways, leading, as confirmed in vitro, to androgen-independent PCa-promoting effects and, in turn, potentially triggering the mechanism relevant for the onset or progression of PCa [137].

In a recent in vitro study [138], both DEHP that DBP have been demonstrated to stimulate the proliferation of PCa cell lines by increasing the expression of cell-cycle-related genes, such as cyclin D1 and the proliferating cell nuclear antigen (PCNA), and decreasing the expression of P21. Moreover, phthalates lead at an increased expression of p-c-fos and p-c-Jun that may promote cell proliferation through transcription factor AP-1 in PCa cell lines. In turn, this caused the activation of the MAPK pathway that might also promote cell proliferation in PCa [138]. In an in vitro PCa cell line model [139], a decreased cellular viability has been demonstrated after treatment with a low concentration of DBP (10^−8^ M). In the same study, the gene expression of ER and AR after treatment with DBP was evaluated, and a reduced expression of ERα was demonstrated, which has been shown to be involved in cell proliferation and carcinogenesis of the prostate [139,140].

Bisphenol A (BPA) is a synthetic polymer compound used predominantly in the production of products made of polycarbonate plastic and epoxy resins that line food and beverage cans [141], and significant levels have been found in the urine of 90% of United States adults [142]. Food contributes to more than 90% of overall BPA exposure while exposure through dust ingestion, dental surgery, and dermal absorption remains below 5% in normal situations [143]. BPA exposure has been associated with a reduced proportion of male births in the populations of several countries, increased the risk of cryptorchidism and hypospadias, and reduced semen quality in males suggesting a possible BPA interference with the male reproductive function [144]. However, the mechanisms underlying the antiandrogen-like effects of BPA remain unclear. Recent data have demonstrated that BPA acts as an AR antagonist interfering with AR-transcriptional activity [145,146]. Moreover, it has been seen that BPA did not impair androgen effects in normal prostate cell lines [147], but acted as an antiandrogen in cancer cells when AR splicing forms were expressed [119]. Thus, androgen signaling seems to be less prone to BPA interference when wild-type AR is expressed, but BPA could interfere with the therapy in patients with advanced PCa via mutant ARs [18]. Indeed, BPA can influence carcinogenesis, modulate PCa cell proliferation, and, for some tumors, stimulate progression [148]. Indeed, a role for BPA and other bisphenols (BPS, BPF) in mechanisms related to androgen-independent PCa-promoting effects [135] as well as to epimutations has been previously shown in a manner similar to phthalates (see above). It is worth noting that in recent years, within the project CLARITY-BPA [97], it has been shown in vivo that chronic low-dose BPA exposure may reprogram adult rat prostate stem cell homeostasis, leading to an increase in stem cell numbers and shifting the lineage commitment toward basal progenitor cells, which may increase carcinogenic risk with aging. The authors of the study conclude that the overall dataset coming from the CLARITY-BPA project provides unbiased evidence on how BPA exposures at human-relevant doses result in adverse effects on the rat prostate gland.

## 7. Conclusions

Since July 2017, the European Commission has indicated that further attention should be given to pesticides/biocides or plasticizer substances for which evaluation or re-evaluation is ongoing or for which confirmatory data have been requested. Too little is still known about many substances, but the potential associations between PCa risk in humans and EDs exposure cannot be underestimated.

## Figures and Tables

**Figure 1 ijms-23-01216-f001:**
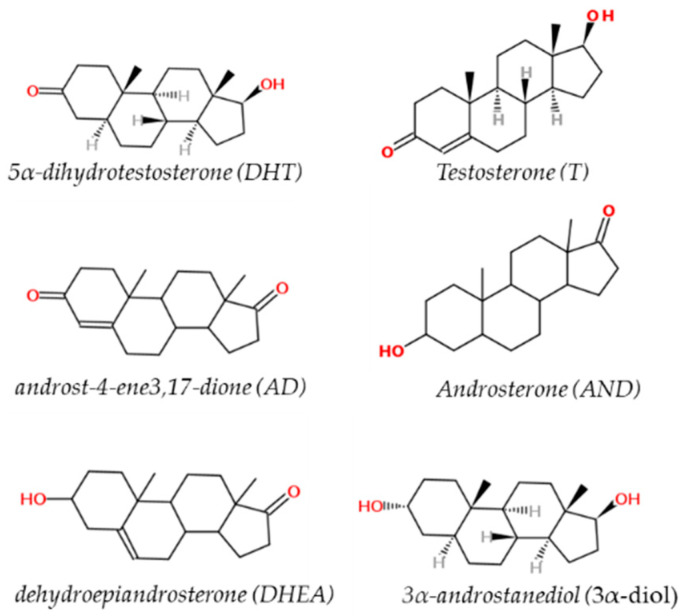
Chemical structure of testosterone (T), 5α-dihydrotestosterone (DHT), androst-4-ene3,17-dione (AD), androsterone (AND), dehydroepiandrosterone (DHEA) and 3α-androstanediol (3α-diol).

**Figure 2 ijms-23-01216-f002:**
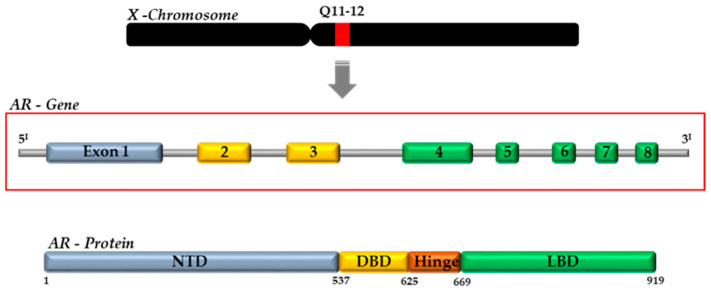
Schematic representation of AR gene and protein, with specific motifs and domains. NTD, N-terminal domain; DBD, DNA-binding domain; LBD, ligand-binding domain; AF, activation function.

**Figure 3 ijms-23-01216-f003:**
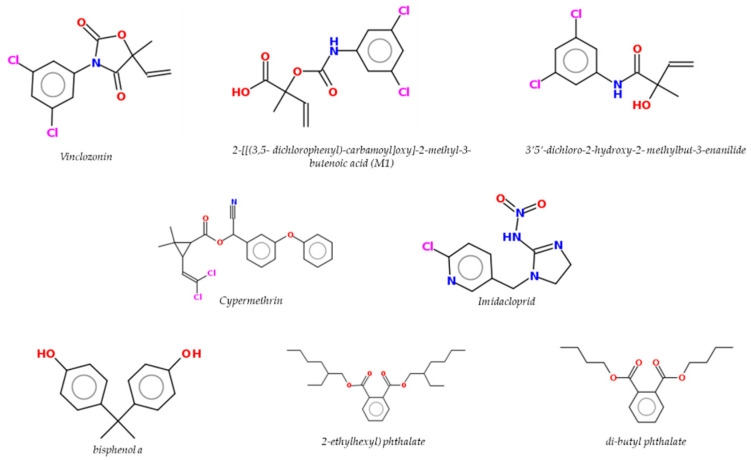
Chemical structures of pesticides/biocides and plasticizers.

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
