# Peer review of "Endocrine Disruptors and Prostate Cancer"

_ijms, 2022, doi:10.3390/ijms23031216_

Round 1
Reviewer 1 Report
Although the work has been greatly improved, I think the association between endocrine disrupters and prostate cancer is still poor. The authors chose a limited category of EDCs but did not accurately describe experimental works demonstrating their influence on the prostate. There is still an extensive description of the role of EDCs on prostate cancer which should be the focus of this review. Most of the references regarding EDCs are old. Below is just a short list of references that could be added to the review. In my opinion, in a review entitled "Endocrine disruptors and prostate cancer", it would be necessary to further investigate the description of the data present in the literature on the association between exposure to EDCs and the onset of this pathology.
Chuang SC, Chen HC, Sun CW, Chen YA, Wang YH, Chiang CJ, Chen CC, Wang SL, Chen CJ, Hsiung CA. Phthalate exposure and prostate cancer in a population-based nested case-control study. Environ Res. 2020 Feb;181:108902. doi: 10.1016/j.envres.2019.108902. Epub 2019 Nov 8. PMID: 31785779.
Wang J, Zhang X, Li Y, Liu Y, Tao L. Exposure to Dibutyl Phthalate and Reproductive-Related Outcomes in Animal Models: Evidence From Rodents Study. Front Physiol. 2021 Dec 8;12:684532. doi: 10.3389/fphys.2021.684532. PMID: 34955869; PMCID: PMC8692859.
Zhang T, Wu J, Zhang X, Zhou X, Wang S, Wang Z. Pharmacophore based in silico study with laboratory verification-environmental explanation of prostate cancer recurrence. Environ Sci Pollut Res Int. 2021 Nov;28(43):61581-61591. doi: 10.1007/s11356-021-14970-8. Epub 2021 Jun 28. PMID: 34184217.
Thorson JLM, Beck D, Ben Maamar M, Nilsson EE, Skinner MK. Ancestral plastics exposure induces transgenerational disease-specific sperm epigenome-wide association biomarkers. Environ Epigenet. 2021 Mar 20;7(1):dvaa023. doi: 10.1093/eep/dvaa023. PMID: 33841921; PMCID: PMC8022921.
Basak S, Das MK, Duttaroy AK. Plastics derived endocrine-disrupting compounds and their effects on early development. Birth Defects Res. 2020 Oct;112(17):1308-1325. doi: 10.1002/bdr2.1741. Epub 2020 Jun 1. PMID: 32476245.
Scarano WR, Bedrat A, Alonso-Costa LG, Aquino AM, Fantinatti B, Justulin LA, Barbisan LF, Freire PP, Flaws JA, Bernardo L. Exposure to an environmentally relevant phthalate mixture during prostate development induces microRNA upregulation and transcriptome modulation in rats. Toxicol Sci. 2019 Jun 14;171(1):84–97. doi: 10.1093/toxsci/kfz141. Epub ahead of print. PMID: 31199487; PMCID: PMC6736208.
Di Lorenzo M, Forte M, Valiante S, Laforgia V, De Falco M. Interference of dibutylphthalate on human prostate cell viability. Ecotoxicol Environ Saf. 2018 Jan;147:565-573. doi: 10.1016/j.ecoenv.2017.09.030. Epub 2017 Sep 15. PMID: 28918339.
de Mello Santos T, da Silveira LTR, Rinaldi JC, Scarano WR, Domeniconi RF. Alterations in prostate morphogenesis in male rat offspring after maternal exposure to Di-n-butyl-phthalate (DBP). Reprod Toxicol. 2017 Apr;69:254-264. doi: 10.1016/j.reprotox.2017.03.010. Epub 2017 Mar 21. PMID: 28341571.
Fang K, Li Y, Zhang Y, Liang S, Li S, Liu D. Comprehensive analysis based in silico study of alternative bisphenols - Environmental explanation of prostate cancer progression. Toxicology. 2022 Jan 15;465:153051. doi: 10.1016/j.tox.2021.153051. Epub 2021 Nov 22. PMID: 34822915.
Wang K, Huang D, Zhou P, Su X, Yang R, Shao C, Wu J. BPA-induced prostatic hyperplasia in vitro is correlated with the unbalanced gene expression of AR and ER in the epithelium and stroma. Toxicol Ind Health. 2021 Oct;37(10):585-593. doi: 10.1177/07482337211042986. Epub 2021 Sep 5. PMID: 34486460.
Salamanca-Fernández E, Rodríguez-Barranco M, Amiano P, Delfrade J, Chirlaque MD, Colorado S, Guevara M, Jimenez A, Arrebola JP, Vela F, Olea N, Agudo A, Sánchez MJ. Bisphenol-A exposure and risk of breast and prostate cancer in the Spanish European Prospective Investigation into Cancer and Nutrition study. Environ Health. 2021 Aug 16;20(1):88. doi: 10.1186/s12940-021-00779-y. PMID: 34399780; PMCID: PMC8369702.
Kaimal A, Al Mansi MH, Dagher JB, Pope C, Varghese MG, Rudi TB, Almond AE, Cagle LA, Beyene HK, Bradford WT, Whisnant BB, Bougouma BDK, Rifai KJ, Chuang YJ, Campbell EJ, Mandal A, MohanKumar PS, MohanKumar SMJ. Prenatal exposure to bisphenols affects pregnancy outcomes and offspring development in rats. Chemosphere. 2021 Aug;276:130118. doi: 10.1016/j.chemosphere.2021.130118. Epub 2021 Feb 26. PMID: 33714148.
Cariati F, Carbone L, Conforti A, Bagnulo F, Peluso SR, Carotenuto C, Buonfantino C, Alviggi E, Alviggi C, Strina I. Bisphenol A-Induced Epigenetic Changes and Its Effects on the Male Reproductive System. Front Endocrinol (Lausanne). 2020 Jul 30;11:453. doi: 10.3389/fendo.2020.00453. PMID: 32849263; PMCID: PMC7406566.
Liu J, Ou C, Zhu X, Tan C, Xiang X, He Y. Potential role of CFTR in bisphenol A-induced malignant transformation of prostate cells via mitochondrial apoptosis. Toxicol Ind Health. 2020 Aug;36(8):531-539. doi: 10.1177/0748233720943750. Epub 2020 Jul 30. PMID: 32729384.
De Falco M, Laforgia V. Combined Effects of Different Endocrine-Disrupting Chemicals (EDCs) on Prostate Gland. Int J Environ Res Public Health. 2021 Sep 16;18(18):9772. doi: 10.3390/ijerph18189772. PMID: 34574693; PMCID: PMC8471191.
Prins GS, Ye SH, Birch L, Zhang X, Cheong A, Lin H, Calderon-Gierszal E, Groen J, Hu WY, Ho SM, van Breemen RB. Prostate Cancer Risk and DNA Methylation Signatures in Aging Rats following Developmental BPA Exposure: A Dose-Response Analysis. Environ Health Perspect. 2017 Jul 11;125(7):077007. doi: 10.1289/EHP1050. PMID: 28728135; PMCID: PMC5744650.
Vandenberg LN, Prins GS, Patisaul HB, Zoeller RT. The Use and Misuse of Historical Controls in Regulatory Toxicology: Lessons from the CLARITY-BPA Study. Endocrinology. 2020 May 1;161(5):bqz014. doi: 10.1210/endocr/bqz014. PMID: 31690949; PMCID: PMC7182062.
Author Response
In agreement with the reviews, the authors investigated in more detail the association between endocrine disruptors and prostate cancer. Moreover, the authors chose only some chemicals of interest in known endocrine interference activity of the prostate gland representative of pesticides, biocides and plasticizers. According to an Italian report on the sales and use of fungicide and neonicotinoids, the vinclozolin and imidacloprid were selected on the basis of their high use. Among pyrethroid molecules, cypermethrin was also selected due to its wide use in agriculture, veterinary, and medical applications as insecticide and biocide. Finally, for their wide use as additives (plasticizers) in the plastics industry, food packaging and food contact materials, DEHP and DBP phthalates and bisphenol have been selected for their well-known characteristics of EDs. The description of the role of EDs on prostate cancer was performed also using the list of references recommended by reviewer. Finally, the references of the manuscript have also been updated with literature, most recent as as recommended by reviewer.

Reviewer 2 Report
The authors have addressed my concerns, therefore, I think the resubmited version can be accepted now.
Author Response
The authors thank the reviewer for considering this new version suitable for publication
Round 2
Reviewer 1 Report
The authors have improved their paper adding more information about EDs and prostate gland.
This manuscript is a resubmission of an earlier submission. The following is a list of the peer review reports and author responses from that submission.
Round 1
Reviewer 1 Report
The review submitted by D. Marcoccia is entitled « Overview: Endocrine disruptors and Prostate ».
General comment:
For a large part (§1-3), this review presents more textbook information (about nuclear receptors, or EDCs or prostate) than really up-to-date and precise data concerning the impact of specific EDCs on prostate functions.
For a future version, the author should focus on the content of §4 and §5 of his current manuscript and develop in more details the molecular and cellular mechanisms of the impact of EDCs (essentially androgenic and anti-androgenic, and potentially others) on the prostate functions and its cancerization.
Specific comments:
The title should be « Endocrine disruptors and Prostate cancer » as most data concern PCa (prostate cancer).
Abbreviations should be introduced at the first occurrence in brackets and then used throughout rather than using the full name/abbreviation again and again.
Numerous examples in current §5 do not concern prostate function. These data are probably interesting but it should be indicated why the author considers they are meaningful for PCa.
Conclusion
In my opinion, this manuscript in its actual state should be rejected from publication in IJMS.

Reviewer 2 Report
The author aims to describe the inflence of endocrine disrupting chemicals on prostate gland. Nevertheless, only a small part of the work is dedicated to deepening this topic. An appropriate classification of EDCs is lacking and the author limits himself only to a superficial description of the action of pesticides / biocides. The literature used is too old and more recent works (such as those of Prins) are not used at all to support the review. There are also several lexical errors that make the understanding of the text unclear. For example, the author misuses the acronym EDCs and often to mean different terms. It doesn't use parentheses but / and this makes reading confusing. The description of the influence of EDCs on the prostate that should have been the main purpose of the review is completely missing.
Reviewer 3 Report
The authors made a review on the endocrine disrupting chemicals and prostate, beNenficial to relevant researchers. The review is necessary. It would be better for the authors to refer more relevant review papers. In the manuscript, actually most of the content deals with the common sense, more summary on latest research would be desired.
- Title: it is too general and it would be better to be more specific
- The description is not very appropriate,’ Endocrine disruption (EDCs) mechanisms in prostate are an overlooked issue’. Actually, it was not overlooked and there are many relevant studies.
- Endocrine disrupting chemicals (EDCs), please replace ‘Endocrine disruption (EDCs)’.
- The first sentence is too long.
- For AR, the full name should be given.
- ‘defined for this reason EDCs’ this is grammar error.
- Lien 23, The XXth century, it is not very proper for such sign.
- Why the authors only summarize the disruption of Pesticides/Biocides, actually there are too many chemicals have androgenic agonistic or antagonistic activity besides of these pesticides/biocides.
- It would be better to discuss more about the structural features on AR LBD, AF2 BF3. There are different of mode of action.
- AR receptors is involved more signal pathways such as Cross talk between androgen and Wnt signaling. It would be better to added relevant description.
- After the last part, the manuscript can be ended with a summary. Something lime future research trends can be briefly mentioned.